



# Seasonal ozone vertical profiles over North America using the AQMEII group of air quality models: model inter-comparison and stratospheric intrusions

Marina Astitha[1], Ioannis Kioutsoukis[2], Ghezae Araya Fisseha[1], Roberto Bianconi[3], Johannes Bieser[4,5], Jesper H. Christensen[6], Owen Cooper[7,8], Stefano Galmarini[9], Christian Hogrefe[10], Ulas Im[6], Bryan Johnson[7], Peng Liu[11], Uarporn Nopmongcol[12], Irina Petropavlovskikh[7], Efisio Solazzo[9], David W. Tarasick[13], Greg Yarwood[12]

[1]University of Connecticut, Civil and Environmental Engineering, Storrs, 06269-3037, CT, United States
[2]University of Patras, Physics Department, 26504, Rio, Greece
[3]Environware srl, via Dante 142, 20863 Concorezzo, Italy
[4]HelmholtzZentrum Geesthacht, Institute of Coastal Research, Geesthacht, Germany
[5]German Aerospace Center(DLR), National Aeronautics and Space Center, Weßling, Germany
[6]Aarhus University, Department of Environmental Science, Frederiksborgvej 399, 4000, Roskilde, Denmark
[7]Cooperative Institute for Research in Environmental Sciences, University of Colorado, Boulder, CO 80309, USA.
[8]Chemical Sciences Division, NOAA Earth System Research Laboratory, Boulder, CO 80305, USA.
[9]European Commission Joint Research Center, Ispra, Italy
[10]Environmental Protection Agency Research Triangle Park, Research Triangle Park, NC, United States
[11]NRC Fellowship Participant at Environmental Protection Agency Research Triangle Park, NC, United States
[12]Ramboll Environ, 773 San Marin Dr., Suite 2115, Novato, CA 94945, USA
[13]Air Quality Research Division, Environment and Climate Change Canada, Downsview, Ontario, Canada.

*Correspondence to*: Marina Astitha (marina.astitha@uconn.edu)

**Abstract.** This study evaluates simulated vertical ozone profiles produced in the framework of the third phase of the Air Quality Model Evaluation International Initiative (AQMEII3) against ozonesonde observations in North America for the year 2010. Four research groups from the United States (U.S.) and Europe have provided ozone vertical profiles to conduct this analysis. Because some of the modeling systems differ in their meteorological drivers, wind speed and temperature are also included in the analysis. In addition to the seasonal ozone profile evaluation for 2010, we also analyze chemically inert tracers designed to track the influence of lateral boundary conditions on simulated ozone profiles within the modeling domain. Finally, cases of stratospheric ozone intrusions during May-June 2010 are investigated by analyzing ozonesonde measurements and the corresponding model simulations at Intercontinental Chemical Transport Experiment Ozonesonde Network Study (IONS) experiment sites in the western United States. The evaluation of the seasonal ozone profiles reveals that at a majority of the stations, ozone mixing ratios are under-estimated in the 1-6 km range. The seasonal change noted in the errors follows the one seen in the variance of ozone mixing ratios, with the majority of the models exhibiting less variability than the observations. The analysis of chemically inert tracers highlights the importance of lateral boundary



conditions up to 250 hPa for the lower tropospheric ozone mixing ratios (0-2 km). Finally, for the stratospheric intrusions, the models are generally able to reproduce the location and timing of most intrusions but underestimate the magnitude of the maximum mixing ratios in the 2-6 km range and overestimate ozone up to the first km possibly due to marine air influences that are not accurately described by the models. The choice of meteorological driver appears to be a greater predictor of model skill in this altitude range than the choice of air quality model.

# 1 Introduction

Since its initiation in 2008, the Air Quality Model Evaluation International Initiative (AQMEII) has brought together scientists from both sides of the North Atlantic Ocean to perform regional model experiments using common boundary conditions, emissions, and model evaluation frameworks with a specific focus on regional modelling domains over Europe and North America (Galmarini and Rao 2011; Rao et al., 2012; Galmarini et al. 2017). Phase 3 of the AQMEII activities (AQMEII3) focuses on joint modelling experiments with the second phase of the Task Force on Hemispheric Transport of Air Pollution (HTAP2) to conduct global and regional assessment of intercontinental transport of air pollutants (Huang et al. 2017; Nopmongcol et al. 2017) and uncertainties stemming from emissions and boundary conditions (Huang et al., 2017; Hogrefe et al. 2017). Investigation of the vertical ozone distribution has occurred during previous phases of the AQMEII activities (Schere et al. 2012, Solazzo et al. 2013) but with model simulations that vary in emissions and boundary conditions for different years. The motivation behind this work is that in AQMEII3, common anthropogenic emission inventories and lateral chemical boundary conditions were implemented by all modeling groups, which helps us further investigate model-to-model variability and performance evaluation.

Regional air quality model evaluation is most commonly performed for ground-level ozone mixing ratios (Hogrefe et al. 2001; Appel et al., 2007, 2012; Herwehe et al. 2011; Solazzo et al., 2012a, b; Im et al., 2015, among others) and less frequently for free tropospheric ozone distributions in longer, non-episodic time frames (Schere et al. 2012; Solazzo et al. 2013; Jonson et al. 2010 using HTAP global modeling systems). This is mainly due to the scarcity of upper-air measurements as well as the need to investigate the efficacy of emissions reduction policies and attainment demonstration which apply to surface ozone exceedances. Nevertheless, accurate representation of the entire troposphere in air quality models influences the prediction of air pollutant vertical distributions, stratosphere-troposphere exchange processes and ground-level mixing ratios. The AQMEII3 framework is ideal for providing the platform and collaborations to assess multi-model simulated ozone vertical profiles from the ground up to the planetary boundary layer and evaluate the models' capability to reproduce ozone mixing ratios aloft as well as to assess contributions from boundary conditions (inert tracer experiments) which have important effects on surface and upper air ozone mixing ratios (Tarasick et al. 2007; Pendlebury et al. 2017).

This study utilizes modeling results for the North American domain from four research groups that participated in AQMEII3 to evaluate seasonal ozone vertical profiles simulated for the year 2010 against ozonesonde observations. The



objectives of this analysis are to: a) evaluate simulated seasonal ozone vertical profiles with ozonesonde measurements; b) assess variations in model performance related to ozone vertical distribution (model inter-comparison), c) assess influence of lateral boundary conditions to ozone profiles within the modeling domain, and d) investigate cases of stratospheric ozone intrusion above the western U.S. during May and June 2010. Because some of the modeling systems differ in their

meteorological drivers, wind speed and temperature are also included in the evaluation. In addition to the ozone profile evaluation for 2010, we analyze chemically inert tracer modeling experiments that estimated the influence of lateral boundary conditions to ozone profiles within the modeling domain. Finally, several cases of stratospheric ozone intrusions are investigated by analyzing ozonesonde measurements and the corresponding model simulations at Intercontinental Chemical Transport Experiment Ozonesonde Network Study (IONS) experiment sites in the western United States (Cooper

et al. 2011; 2012). IONS-2010 was a component of the CalNex (Research at the Nexus of Air Quality and Climate Change) 2010 experiment, which focused on understanding the effects of air pollutants on air quality across California (Ryerson et al., 2013). The data and methods of analysis are described in Section 2; Evaluation and model inter-comparison of ozone seasonal profiles are provided in Section 3; Results from the model experiments using chemically inert tracers are provided in Section 4 and the case study of stratospheric ozone intrusions is discussed in Section 5. The summary and conclusions are

presented in Section 6.

## 2 Data and Methods

### 2.1 Atmospheric Modeling Systems

The base case simulations are used in this study are performed by all AQMEII3 participants using lateral chemical boundary conditions are prepared from global concentration fields simulated by ECMWF's global chemistry model C-IFS

(Flemming et al., 2015). Table 1 provides an overview of each participating research group/Institution, their modeling systems and main specifications of the simulations. A detailed description of the four modeling systems (US1, US3, DE1 and DK1) is provided in Solazzo et al. (2017). Harmonization of all model simulations is achieved by specifying a common simulation time period (January - December 2010), common regional anthropogenic and fire emission inventories (Pouliot et al., 2015), and common lateral chemical boundary conditions. The 2008 National Emission Inventory is used as basis for the

2010 emissions with necessary updates described in (Pouliot et al., 2015). Anthropogenic emissions totals are the same for all models, but each group uses their own system to spatially disaggregate and temporally allocate emissions to their gridded domain (for example: DE1 and DK1 use HTAP emissions while US3 and US1 use the Sparse Matrix Operator Kernel Emissions (SMOKE); SMOKE emissions were provided on hourly basis while HTAP is monthly, so the temporal, vertical and chemical distributions might be different among models). The simulations differ in the modeling systems (air quality and

meteorology), horizontal and vertical grid spacing, chemistry modules and deposition schemes as well as biogenic emissions. Each modeling group was free to use the meteorological model of their choice based on compatibility with their chemical transport model. More details on the AQMEII3 modeling experiments are included in the technical note by Galmarini et al.



(2017). All research groups interpolated their results into the same 0.25x0.25 degree grid spacing before submitting the model outputs to the common data platform for the analysis (Joint Research Institute's ENSEMBLE system).

Each modeling group also included three non-reactive tracers in their simulations (Table 1). These tracers are designed to track the inflow of ozone from the lateral domain boundaries and are specified as lateral boundary conditions, with no emissions or chemical formation/destruction occurring within the modeling domain. All tracers undergo advection, diffusion, cloud mixing/transport, scavenging, and deposition, but no chemistry. The tracer mixing ratios and their vertical profiles are used to investigate the sensitivity of ozone to the lateral boundary conditions. It should be noted that these inert tracers were not intended to provide a quantitative attribution of ground level ozone to ozone boundary conditions. As noted by Baker et al. (2015) and Nopmongcol et al. (2017), inert tracers would overestimate such contributions due to the lack of chemical loss terms which are considered in other attribution tools such as reactive tracers or ozone source apportionment. However, using them in a relative manner helps identify the sensitivity of modeled ozone mixing ratios to lateral boundary conditions. The definition of each tracer is as follows:

1) BC1: For layers below 750 hPa (~2.5 km), the boundary conditions for this tracer are set to the same C-IFS ozone mixing ratios used as ozone boundary conditions for the regional models. For layers above 750 hPa, the boundary conditions for this tracer are set to zero.

2) BC2: For layers between 750 hPa (~2.5 km) and 250 hPa (~10 km), the boundary conditions for this tracer are set to the same C-IFS ozone mixing ratios used as ozone boundary conditions for the regional models. For layers below 750 hPa and above 250 hPa, the boundary conditions are set to zero.

3) BC3: For layers above 250 hPa (~10 km), the boundary conditions for this tracer are set to the same C-IFS ozone mixing ratios used as ozone boundary conditions for the regional models. For layers below 250 hPa, the boundary conditions are set to zero.

**2.2 Ozonesonde sites and statistical metrics**

Ozonesondes are obtained from various networks with data availability for the year 2010. Ten sites across North America are selected for seasonal and annual analyses (Fig. 1a) and five additional sites located in the western U.S. (Fig. 1b) are selected for studies of stratosphere / troposphere exchange (note that the Trinidad Head site was selected for both types of analyses and is shown in both Fig. 1a and Fig. 1b). Information on data networks and station characteristics, including the number of launches available for analysis, are summarized in Table 2. The modeled and observed ozone fields were interpolated at the following eighteen (18) standard vertical heights above ground level (m): 0, 100, 250, 500, 750, 1000, 1500, 2000, 3000, 4000, 5000, 6000, 7500, 8500, 10000, 12000, 15000, 18000. The ten sites depicted in Figure 1a had launches throughout the entire year and are used to construct seasonal average profiles by averaging over all available launches in a given season at each vertical height. Seasonal averages are chosen to evaluate how models capture transport and photochemistry processes that influence ozone formation (Winter: DJF; Spring: MAM; Summer: JJA; Fall: SON). The modeled ozone mixing ratios are sampled in accordance to the available ozonosondes, thus the seasonality of the vertical



ozone profiles is under-represented since the ozonesondes are not continuous throughout each month (Lin et al. 2015). The evaluation of ozone vertical profiles is performed for layers up to 8.5km since there is less confidence on the tropopause placement for the regional models which was evident by large errors in ozone mixing ratios above 8.5 km (not shown). The study by Makar et al. (2010) has shown that when models predict a tropopause height above the one implicit in the ozone background conditions (ozone climatology), then higher ozone mixing ratios will become available in the upper troposphere resulting in high model errors.

IONS experiments are aimed at measuring tropospheric ozone variability across North America (Thompson et al., 2007). During the IONS-2010 experiment, ozonesondes were launched almost daily between May 10 and June 19, 2010. Its main goal was to determine the latitudinal variability of baseline ozone along the California coast from the surface to the tropopause (Cooper et al. 2011). A total of 230 ozonesondes were launched at seven sites, one in southern British Columbia and six in California. Figure 1b shows the locations of the six IONS ozonesonde sites in California. All IONS sites are located in very rural areas far from fresh emissions. Four of the sites are right on the coast, almost in the water, (TH, RY, PS, SN) and in the lowest few hundred meters of the atmosphere they represent depleted ozone from the marine boundary layer, while the other two are inland (SH, JT).

The statistical metrics used in the model evaluation and model inter-comparison are root mean square error (RMSE), Pearson correlation coefficient (R), 95% bootstrapping confidence intervals (indicates significance in differences between models and observations), and the Fractional Difference indicator (FD) used in the stratospheric intrusion case study only, defined as follows:

$$FD(\%) = 200 \, (mod-obs)/(mod+obs)$$

where *mod* and *obs* denote the modeled and observed ozone values. If all modelled values lie within a factor of 2 of the observations then FD is between -66.7% to +66.7%, and if all modelled values lie within a factor of 3 of the observations then FD is between -100% and +100%. The interpretation of the results is made with caution due to the incommensurability of the comparison of point measurements with grid cell model values.

## 3 Evaluation and model inter-comparison of ozone seasonal profiles for 2010

The ozone vertical profiles for each season and station (Fig. 2-4 and boxplots in Fig. S1 in the supplement) highlight the variability of model behavior depending on the specific model configuration as well as the impact of seasonal cycles that alter emissions, transport and transformation of ozone. During winter, all models under-estimate the mean and variability of ozone mixing ratios in the 1.5-5 km vertical levels for all stations, with the exception of Boulder, Narragansett and Huntsville. In most cases, the 95% bootstrapping confidence intervals do not overlap between models and observations in the 1.5 to 5 km height range, indicating that the differences in the mean are statistically significant. Model behavior near the surface (0-1 km) varies, with the majority of the models agreeing with observations. There is a notable tendency for most models to underestimate the 0-1 km mean ozone mixing ratios for the two easternmost sites (Yarmouth and Narrangasett;





Fig. 3). The ozone mixing ratios exhibit larger variability in the upper layers (5-8.5 km) with the models behaving differently depending on the site and altitude.

During spring, all models show better performance for the lower layers for most stations. Variable behavior is shown in the two easternmost sites (Yarmouth and Narrangasett; Fig. 3). In Yarmouth, the observed ozone is underestimated by all models in the 0.75-6 km range while the models agree with observations in the lower layers. At Narrangansett, a similar underestimation is noted in the 2-6 km range but the models' behavior varies in the lower layers. The results for Narrangansett must be viewed with caution due to the limited number of profiles, which varies from 5 to 8 for each season.

During summer, all models over-predict ozone in the 0-0.5 km layer at the northern sites of Bratts Lake and Stony Plain. For the Egbert site, DK1 shows a significant over-prediction in the 0-2 km range, which might be influenced by the model's coarse grid spacing (50 km; Table 1) as opposed to the other modeling systems. Egbert is located near the Great Lakes (Fig. 1a, STN456) and the complexity of the geography might not be resolved adequately at the specific horizontal resolution. The relatively coarse grid spacing used by DK1 might also explain the similar behavior at Wallops Island where DK1 results stand out from other models in the lowest 0-2 km, possibly resulting from a different representation of the land/water interface and resulting mixing heights. However, as noted below, the summer temperature profiles for DK1 shown in Figure S2 do not offer conclusive evidence that the ozone differences can be attributed to differences in mixing due to different grid spacing. All models, except DK1, overpredict the mean ozone mixing ratios Narrangansett (eastern part of the domain) at 0-0.25 km and the same behavior is seen in Yarmouth. At the westernmost site, Trinidad Head, all models overpredict ozone in the 0-1 km range. Finally, the mean ozone profiles during fall are generally well represented by all models with some variations depending on the site and height, which cannot be generalized. One common pattern for the eastern and northern sites is the under-prediction of ozone in the 3-6 km range (the exception is Wallops Island; SON profiles are shown in the supplemental material, Fig. S1).

By evaluating the error in the seasonal ozone vertical profiles for two height ranges (lower troposphere (LT; 0-2 km) and upper troposphere (UT; 2-8.5 km)), we observe the expected error magnitude difference between LT and UT given the increase in the ozone mixing ratios in the upper layers (Fig. 5). For this analysis, the RMSE is calculated at each of the standard altitude levels listed in Section 2.2 using all available launches in a given season and then averaged across all standard levels in the LT and UT ranges. The LT errors are 2-4 ppb higher for the summer compared to other seasons for most models (the average RMSE for all stations and models during summer is 12 ppb and 10 ppb for the fall). The lowest LT errors are seen in winter and spring with an average error of ~8 ppb across all models and sites. At most sites, the DK1 simulations for LT exhibit a higher RMSE than other models during summer and fall with RMSE values that range from 6 to 32 ppb (32 ppb RMSE for the Wallops Island site and 24 ppb for Huntsville in the fall are the maximum values). Vertical profiles of temperature and wind speed for DK1 do not show large variations for Wallops Island during summer (Fig. S2, S3), but for Huntsville the temperature profile is underestimated consistently for all seasons and layers (Fig. S2). Wind speed profiles were not available for Huntsville.



There is a peak in the LT and UT RMSE at Yarmouth during fall associated with all modeling systems. Since this is the easternmost site in the model domain, it might indicate that the eastern boundary condition is not appropriate for the fall or the weather variables exhibit errors that influence ozone mixing ratios. The temperature profiles are very similar between all models and observations for Yarmouth (Fig. S2), but the LT wind speed is underestimated by DE1 and US1 (Fig. S3). The

wind and temperature profiles for US3 in Yarmouth in the fall do not show any significant variation from the observations to explain the higher RMSE value. In general, the average RMSE over all stations for the LT increases for all models in the following order: winter, spring, fall, summer.  All models have similar error magnitudes for the LT, with DK1 being an outlier during summer and spring when it has noticeably higher RMSE values than the other models. The seasonal change in the variance of simulated and observed LT ozone mixing ratios is the same with the change seen in the RMSE values (higher

during summer and fall and lower during spring and winter). All models are less variable than the observations with the exception of DK1 for summer and fall.

For the UT, the highest errors occur during winter and spring. The average RMSE across all stations and models during spring is 33 ppb; 26 ppb for winter; 22 ppb for summer and 15 ppb for fall. There is a tendency for all models to produce high UT errors for the Boulder site during winter and spring and for Huntsville and Trinidad for spring. For Trinidad and

Huntsville, only DK1 underestimates the observed temperature for all vertical levels and seasons, whereas it overestimates the UT temperature profiles for Boulder (Fig. S2). These results do not provide any insights into the cause of the common high UT errors across all models but given that they occur in all models despite different meteorological drivers and model configurations they do suggest that the lateral boundary conditions are a major factor. In general, the average UT RMSE over all stations increases for all models in the following order: fall, summer, winter, spring. The higher UT errors agree with

the vertical profile analysis discussed previously, where large deviations from the observed ozone profiles is seen at the 1-6 km vertical range. The seasonal change in the variance of simulated and observed UT ozone mixing ratios is the same with the change seen in the RMSE values (higher during spring and winter and lower during summer and fall). All models are less variable than observations with the exception of DK1 for winter and summer.

The statistical evaluation and inter-comparison of modeled ozone profiles for the lower (0-2 km) and upper troposphere

(2-8.5 km) are further explored with the Taylor diagrams in Fig. 6 for each season and vertical range. For these Taylor diagrams, observations and model results for each standard vertical level were averaged over all vertical levels in a given vertical range (LT or UT) for each launch and the resulting vertical averages for each launch were then used to compute the metrics depicted in the diagrams. Thus, the variability metrics (correlation coefficient and normalized standard deviation) measure the temporal variability across launches in a given season at a given station. The seasonal LT Taylor charts

highlight the variability in model performance during all seasons. One common feature throughout all seasons is that most models underestimate the observed variability at most sites as indicated by standard deviation ratios (measured by concentric circles around the origin) of less than 1. During winter (Fig. 6, DJF_LT) very low (and negative) correlations and high centered RMS differences are evident for the western sites of Trinidad Head and Kelowna (all models) in the LT. The





predictions are improved for Egbert where all models have correlations above 0.85 and low RMSE. In general, LT variations at both sites in the western part of the domain are not captured well by the four modeling systems during all seasons.

Spatial variability in LT model performance is still evident in the statistical metrics for spring (Fig. 6, MAM_LT). LT correlations are somewhat improved for the summer (with 13 points showing correlations above 0.6) and further improved in the fall with most of the points having correlations above 0.6. It is apparent that no single model outperforms the others in the station-by-station comparison. When considering the overall statistics for all stations (Fig. S4), US3, US1 and DE1 share similar performance for spring, summer and fall. It is interesting to also note the differences/commonalities between the models: US3 and US1 share common meteorological inputs, while US3 and DE1 are based on the same air quality model (though a different model version). There is no obvious attribution of the model performance to these differences and commonalities when looking at each individual station.

As discussed earlier, the UT ozone mixing ratios are more challenging for all four modeling systems and this is evident by looking at the station-based Taylor diagrams (Fig. 6, UT) as well as the station-averaged diagrams in the supplementary material (Fig. S4). As was the case for the LT, the modeled temporal variability tends to be lower than the observed temporal variability across all models and sites. Models US1 and US3 have very similar performance at most stations. During summer and fall, there is less spread in the model results with US3, US1 and DE1 performing similarly for most stations and DK1 having the most distinct behavior compared to the other three models. For example, DK1 at Wallops Island during summer and fall has high RMSE values (shown in Fig. 5) and we can see from Fig. 6 (JJA_UT and SON_UT, red triangle) that the correlation is low and RMSD is high.

The variability of model performance and the lower correlations during winter are further explored by analyzing the profiles for winter only and for each region separately. The average of winter ozone profiles over all stations (Fig. 7a) shows under-prediction in the 1-6 km height range. This common condition is also seen for the Western, Northern and Eastern sites separately (Figs. 7 b-d). For the Eastern sites, ozone is under-predicted from the surface to 6 km, while for the Western sites all models indicate over-prediction of ozone in the levels below 250 m. To gain insight into how lateral boundary conditions might have influenced the performance of three of the modeling systems (DE1, US3, and DK1), the chemically-inert tracer results are discussed in the following section for all seasons and sites.

## 4 Influence of lateral boundary conditions to ozone profiles using chemically-inert tracers

Three chemically-inert tracers are included with the simulations by all modeling groups but only three of the modeling systems provided 3D data of the tracer mixing ratios (Table 1). As described in Section 2.1, these tracers undergo advection, diffusion, cloud mixing/transport, scavenging, and deposition but no chemistry. We are interested in the relative contribution of each lateral boundary tracer to the total tracer mixing ratios and the characteristics of each tracer's vertical profile at the ten ozonesonde sites. The relative contribution of each tracer (BC1, BC2 and BC3) is assessed by normalizing each one with the sum of all tracer mixing ratios (BCtot=BC1+BC2+BC3). This normalization allows us to compare contributions from each tracer at each site and season (Fig. 8). The normalized values are assessed for three vertical layers: LT represents the





lower troposphere (0-2km); MT the middle troposphere (2-8.5 km) and UTLS the upper troposphere to lower stratosphere (8.5-18 km) following Nopmongcol et al. (2017). BCtot is calculated for each vertical layer separately. More specifically, the percentage contribution from each tracer BC1, BC2 and BC3 to the LT, MT and UTLS for each model, station and season is analyzed and discussed.

5      The lower troposphere mixing ratios (LT) is influenced by both BC1 (lateral boundary set to non-zero below 750 mb) and BC2 (lateral boundary set to non-zero between 750 and 250 mb). The relative contributions of BC1 and BC2 depend on season and station location. For example, during summer, BC2 contribution is stronger for all sites (50-85%) except Trinidad Head where BC1 and BC2 have an almost equal contribution. This indicates the importance of lateral boundary conditions up to 250mb for the lower troposphere ozone mixing ratios (0-2km). Looking back at the poor model performance for the western sites of Trinidad Head and Kelowna for winter and summer (Fig. 6; DJF_LT and JJA_LT), one possible explanation and point of further investigation would be the influence of lateral boundary conditions up to 10 km (250 mb).

The MT tracer mixing ratios is primarily influenced by the BC2 tracer with some contribution from BC3. The BC3 contribution to MT is more pronounced for the DE1 model for all seasons and sites. The US3 model shows a small contribution to MT from BC1 and BC3, except for Boulder and Huntsville. This means that the lateral boundary conditions within the vertical range 750-250 mb primarily influence the ozone mixing ratios in the MT. The UTLS mixing ratio is almost exclusively influenced by the BC3 tracer for all seasons, models and sites.

Since chemistry is not part of the BC experiments, the relative contributions analyzed here are primarily proxies for the transport and deposition mechanisms. The seasonality of contributions seen in the LT and MT layers is, thus, directly related to planetary boundary layer (PBL) processes and designates the significance of the influence that lateral boundary conditions have during each season. An in-depth multi-model comparison of the inert tracer mixing ratios at the surface is provided by Liu et al. (2017).

## 5 Case study: stratospheric intrusions during May-June 2010

Stratosphere to troposphere transport is an important process that affects tropospheric ozone (Stohl et al., 2003). This analysis addresses the ability of different air quality modeling systems to represent the relevant dynamical processes during springtime stratospheric intrusions above the western U.S. capitalizing on the AQMEII3 simulations for 2010 and ozonesondes from the IONS campaign (Cooper et al. 2011; 2012). For average conditions, the upper tropospheric ozone mixing ratios decrease from north to south for a given altitude (Liu et al., 2013). The IONS measurement data demonstrate a gradient of ~40 ppb at 8 km a.s.l. between the northernmost and southernmost coastal sites during the study period (Fig. 9a). Factors contributing to the gradient include stronger influence from a lower tropopause and more frequent stratospheric intrusions at higher latitudes, as well as greater influence from low-ozone tropical air masses at lower latitudes (Cooper et al., 2011). Below 4 km there is little latitudinal difference in the average ozone profiles. Only JT (Joshua Tree; Fig. 1b), downwind of the Los Angeles Basin, exhibits a departure from the mean profile with enhanced mixing ratios (Fig. 9a).





A comparison of the distribution of modeled versus observed ozone profiles (5th, 50th and 95th percentiles using 131 profiles at 6 IONS sites; Fig. 9b) reveals that the median ozone mixing ratio increases with altitude in the first 1000 m, as deposition reduces ozone mixing ratios near the ground (e.g. Chevalier et al., 2007). In addition, the coastal sites (four out of six) represent depleted ozone from the marine boundary layer, which can also be seen in the mean ozone profiles for each

station in Fig. 9a; the four coastal sites have almost identical ozone mixing ratios between 0 to 250 m. The models might not be able to capture the influence of marine air due to the horizontal grid spacing and how each model treats subgrid scale processes (i.e. for a grid cell that includes both land and sea surface). The effect of surface processes on ozone is also evident by the strong gradient in the first 2 km of the troposphere, ranging between 10 and 20 ppb $km^{-1}$ at all sites. The observed and modeled median profiles are in close agreement mostly above 250 m (Fig. 9b). All models show a similar general structure,

with overestimation of the median in the first km and with few exceptions above 6km. Another common feature to all models is the smaller range between the 5th and 95th percentiles compared to the observed spread at all levels, with the only exception being DK1 in the first 2 km. The positive bias in the PBL during summer at North American stations was also found for the simulations performed as part of AQMEII Phase 1 (e.g., Solazzo et al., 2013) although it should be noted that those simulations were performed with a different suite of models for a different year, were driven by different boundary

conditions, and were not evaluated at the IONS locations. In the 1st km, the overestimations are likely due to inaccuracies in PBL processes such as marine air influence, emissions, photochemistry as well as deposition. Given the proximity of the IONS sites to the regional domain boundaries, the analysis of the inert boundary tracers in Section 4, and the comparison of global and regional model simulations at Trinidad Head presented in Hogrefe et al. (2017), the errors above 6 km are likely caused by errors in the representation of tropopause dynamics in the models that affected the downward mixing of higher

stratospheric ozone mixing ratios.

The identification of stratospheric intrusions is typically quantified using tracers of stratospheric origin in numerical models. On this basis, seven stratospheric $O_3$ intrusions occurred in the western U.S. during the IONS2010 campaign in May-June 2010 (Cooper et al., 2011; Lin et al., 2012a, b). The four strongest intrusions occurred on May 22–24, May 27–29, June 7–8, and June 9–14 (Lin et al. 2012a, b). Enhanced ozone mixing ratios in combination with very low relative humidity

(RH) provides a qualitative proxy for dry air of possible stratospheric origin. High isentropic potential vorticity (IPV) in the troposphere and high total ozone column (TOC) are other indicators of stratospheric air and tropopause folding. Figure 10 displays both IPV at 330K and TOC fields over the western U.S. during 28 May and 10 June, when the strongest stratospheric intrusions occurred (source: ERA-interim). Both fields demonstrate higher than normal values over the region during the examined periods. This result is also supported from the soundings at the six IONS sites (Figure S5). Dry air

masses with enhanced $O_3$ are recorded at various levels, in spatial agreement with areas of enhanced TOC and IPV (Figure 10). May 28 and June 8-9, 2010 are selected as the most representative of strong stratospheric intrusions and the vertical ozone profiles for all models and stations are depicted in Fig. 11. On May 28, the soundings show high ozone values (above 100-150 ppb) for the northern sites (TH, RY and SH) in the 6-10 km range, and for the southern sites (PS, SN, JT) in the 2-5 km range; these high ozone values coincide with a strong drop in RH. The high ozone mixing ratios are not captured by any



model, except at Trinidad Head (TH) and Shasta (SH). Similar performance is seen in the June 9 vertical profiles, where the models capture the vertical gradient of the ozone mixing ratios but not the high values seen in the northern sites, RY and PS (all vertical profiles are included in the supplementary documentation, Fig. S5).

We also calculated the aggregated Fractional Difference indicator (FD) across all stations. The general model errors found earlier, such as the tendency for all models to overestimate mixing ratios in the 1$^{st}$ km, are also evident in the FD plot (Figure 12). Moreover, the tendency of some models to depart from the average error profile is also reproduced, such as the underestimation of DE1 between 1-2 km and the overestimations of DK1 in the 5-7 km layer. When calculating the FD at each site, it is found that the overestimation in the 1$^{st}$ km occurs at all sites and has a latitudinal gradient across the coastal sites with larger values towards the south, which relates to the impact of the marine boundary layer. Above 5 km, the bias also has a latitudinal gradient starting with negative values in the north (TH) and progressively becoming positive moving southwards. During episodic conditions, significant over-estimations and under-estimations are evident above 9 km at some sites (e.g., RY in panel b, PS in panel d). Those high FD values of both signs are found at the sites exhibiting stratospheric intrusion signals in Figure S5 (e.g., RY at May 27, PS at June 11), indicating that the stratosphere-troposphere exchange in the regional model and/or the C-IFS model providing boundary conditions may not be fully captured during these episodes. The performance of the modeling systems appears to be more closely linked to the meteorological driver rather than the actual air quality model. The two simulations using CMAQ (US3 and DE1) do not produce similar results at any of the sites, although they share the same BCs and emissions. In contrast, the CMAQ and CAMx simulations (US3 and US1 respectively) which share common meteorological fields, and thus the same PBL scheme (but use a different vertical resolution as noted by Liu et al., 2017) have rather similar results.

## 6 Conclusions

This study analyzes four annual air quality model simulations for North America performed under AQMEII3 to evaluate seasonal ozone vertical profiles for the year 2010 against ozonesonde observations. The objectives of this analysis are to: a) evaluate simulated seasonal ozone vertical profiles with ozonesonde measurements; b) assess variations in model performance related to ozone vertical distribution (model inter-comparison), c) assess the influence of lateral boundary conditions on ozone profiles within the modeling domain, and d) investigate cases of stratospheric ozone intrusions in the western U.S. during May-June 2010.

The evaluation of the seasonal ozone profiles reveals that at a majority of the stations, ozone mixing ratios are under-estimated in the 1-6 km range. Model performance as measured by RMSE is better during winter and spring for the lower troposphere (LT, 0-2 km) and during summer and fall for the upper troposphere (UT; 2-8.5 km). In general, the average RMSE over all stations for the LT increases for all models in the following order: winter, spring, fall, summer. Average RMSE for all stations and models during summer is 12 ppb, 10 ppb for the fall, and 8 ppb for winter and spring. Average RMSE for all stations for the UT during spring is 33 ppb; 26 ppb for winter; 22 ppb for summer and 15 ppb for fall. There is





a tendency for all models to agree on high UT errors for the Boulder site during winter and spring and for Huntsville and Trinidad Head during spring. For both LT and UT, the same seasonal change noted in the RMSE is seen in the variance of ozone mixing ratios for both observations and model results, with the majority of the models exhibiting less variability than the observations.

The chemically-inert tracers provide a relative assessment of influences of the lateral boundary conditions on ozone profiles. The results indicate that the lower troposphere mixing ratios (LT) are influenced by both BC1 (lateral boundary set to non-zero below 750 hPa) and BC2 (lateral boundary set to non-zero between 750 and 250 hPa). The relative contributions of BC1 and BC2 depend on season and station location, with the BC2 contribution being stronger in the summer for all sites (50-85%) compared to BC1. This highlights the importance of lateral boundary conditions up to 250 hPa for lower

tropospheric ozone mixing ratios (0-2 km). The Middle Troposphere mixing ratios (MT) are primarily influenced by the BC2 tracer with some contribution from BC3 (lateral boundary set to non-zero above 250 hPa). The Upper Troposphere-Lower Stratosphere mixing ratios (UTLS) are almost exclusively influenced by the BC3 tracer for all seasons, models and sites.

For the stratospheric intrusion case study, the comparison of the four modeling systems against $O_3$ soundings in California during May-June 2010 revealed that the models can reproduce the location and timing of most intrusions but

underestimate the magnitude of the maximum mixing ratios in the 2-6 km range. There is a general tendency of the models to overestimate ozone mixing ratios in the 1 km layer adjacent to the surface and above 5 km. The former is possibly related to inaccuracies in surface and/or PBL processes while the latter points to potential errors in boundary conditions and/or the representation of the exchange between the upper troposphere and the lower stratosphere in the regional models. The differences between the four modeling systems are mostly evident above 6 km and the choice of meteorological driver

appears to be a greater predictor of model skill in this altitude range than the choice of air quality model.

**Acknowledgments**

We gratefully acknowledge the contribution of all research groups and organizations that provided the datasets used in this study: US EPA (North American emissions processing and gridded meteorology); ECMWF/MACC (Chemical boundary conditions); the WMO World Ozone and Ultraviolet Data Centre (WOUDC) and its data-contributing agencies provided

North American ozonesonde profiles; additional ozonesonde profiles were downloaded from NOAA's Earth System Research Laboratory, Global Monitoring Division (https://www.esrl.noaa.gov/).

The modeling and observational data generated for the AQMEII phase 3 are accessible through the ENSEMBLE data platform (http://ensemble.jrc.ec.europa.eu) upon contact with the managing organizations. Joint Research Center Ispra/Institute for Environment and Sustainability provided its ENSEMBLE system for model output harmonization and

analyses and evaluation.

The views expressed in this article are those of the authors and do not necessarily represent the views or policies of the U.S. Environmental Protection Agency.

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



**Table 1:** Specifications of the modeling systems used in this study. All models use chemical boundary conditions from C-IFS (see notes).

| Institution | Abbreviation | Modeling Systems | Boundary Conditions (meteo) | Horizontal Grid spacing | Vertical Layers | Approximate height at 1st layer | Inert tracers |
|---|---|---|---|---|---|---|---|
| **U.S. EPA** | US3 | WRF3.4/ CMAQ5.0.2 | NCEP | 12km | 35 layers up to 50hPa | 19 m | Yes |
| **Helmholtz-Zentrum Geesthacht (Germany)** | DE1 | COSMO-CLM/ CMAQ5.0.1 | NCEP | 24km | 30 layers up to 50hPa | 40 m | Yes |
| **RAMBOLL Environ (U.S.)** | US1 | WRF3.4/ CAMx6.2 | NCEP | 12km | 26 layers up to 97.5hPa | 19 m | No |
| **Aarhus University (Denmark)** | DK1 | WRF/ DEHM | ECMWF | 50km | 29 layers up to 100hPa | 25 m | Yes |

Notes: C-IFS=ECMWF's Composition Integrated Forecasting System (IFS); US3 and US1 use the WRF model with ACM2
5   PBL module (Asymmetric Convective Model with nonlocal upward mixing and local downward mixing (Pleim, 2007)).
DK1 uses MYJ PBL scheme in WRF: Mellor–Yamada–Janjic (Janjic, 1994).





**Table 2:** Names, codes and geographic locations of ozonesonde sites. Next to the code is a characterization of the site location relative to the model domain. The elevation at these sites ranges from sea level to 1.6 km above sea level.

| ID | CODE | NAME | LON | LAT | NETWORK | Number of profiles |
|----|------|------|-----|-----|---------|--------------------|
| 1 | STN021 / North | Stony Plain | -114.1 | 53.54 | ECCC | 43 |
| 2 | STN107 / East | Wallops Island | -75.47 | 37.93 | NASA-WFF | 53 |
| 3 | STN338 / North | Bratts Lake | -104.7 | 50.20 | ECCC | 49 |
| 4 | STN418 / South | Huntsville | -86.64 | 34.72 | NOAA-ESRL | 51 |
| 5 | STN445 / West | Trinidad Head | -124.16 | 40.80 | NOAA-ESRL | 77 |
| 6 | STN456 / North | Egbert | -79.78 | 44.23 | ECCC | 54 |
| 7 | STN457 / West | Kelowna | -119.4 | 49.94 | ECCC | 74 |
| 8 | STN458 / East | Yarmouth | -66.1 | 43.87 | ECCC | 70 |
| 9 | STN487 / East | Narragansett | -71.42 | 41.49 | NOAA-ESRL | 26 |
| 10 | BOULDER/Central | Boulder | -105.25 | 40.00 | NOAA-ESRL | 44 |
| 11 | RY | Point Reyes | −122.95 | 38.09 | IONS2010 | 32 |
| 12 | PS | Point Sur | −121.89 | 36.30 | IONS2010 | 36 |
| 13 | SN | San Nicolas Island | −119.49 | 33.26 | IONS2010 | 23 |
| 14 | JT | Joshua Tree | −116.39 | 34.08 | IONS2010 | 36 |
| 15 | SH | Shasta | −122.49 | 40.60 | IONS2010 | 33 |

**Notes:** NOAA/ESRL: National Oceanic and Atmospheric Administration/ Earth System Research Laboratory (data downloaded from https://www.esrl.noaa.gov/, May 2016); NASA/WFF: National Aeronautic and Space Agency/Wallops Flight Facility; ECCC: Environment and Climate Change Canada; IONS: Intercontinental Chemical Transport Experiment Ozonesonde Network Study. Data from ECCC and NASA-WFF were downloaded from the WMO World Ozone and Ultraviolet Data Centre (WOUDC; doi:10.14287/10000001).



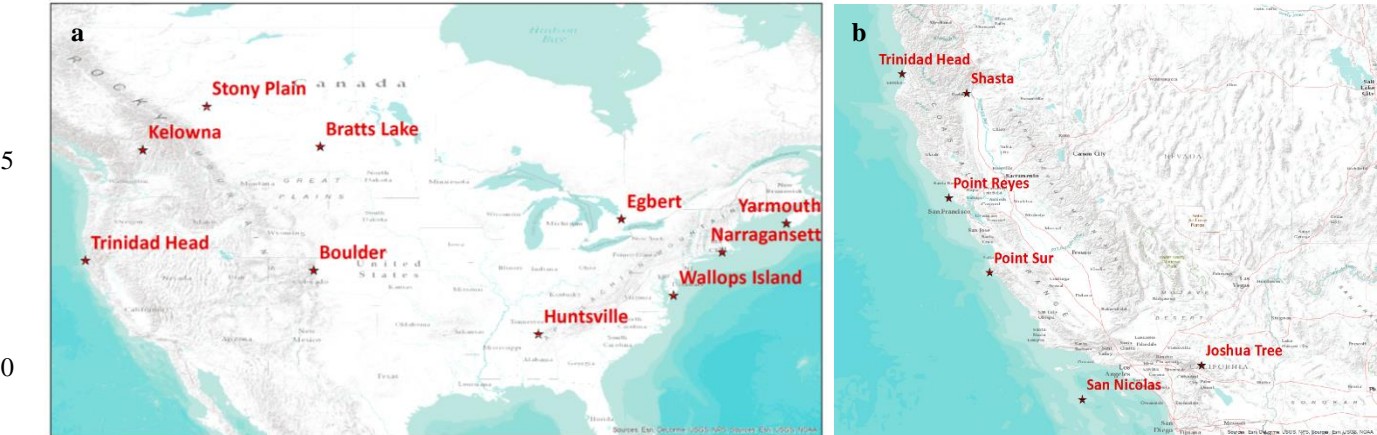

**Figure 1:** Geographic maps of ozonesonde monitoring sites for 2010: a) North America (seasonal analysis) and b) western U.S. (stratospheric intrusion evaluation).





**Figure 2:** Seasonal vertical profiles of ozone mixing ratios for 2010, for three stations located in the northern part of the domain. The horizontal lines indicate the 95% bootstrapped confidence interval for each vertical layer. Note: Bratts Lake has only four ozonesondes for SON and Stony Plain does not include model outputs from DE1 as the model domain does not cover that station.





**Figure 3:** Seasonal vertical profiles of ozone mixing ratios for 2010, for three stations located in the eastern part of the domain. The horizontal lines indicate the 95% bootstrapped confidence interval for each vertical layer. Note: Narrangasett has limited amount of ozonesondes for all seasons (less than 10 for each season) and the results should be viewed with caution.



**Figure 4:** Seasonal vertical profiles of ozone mixing ratios for 2010, for three stations located in the central (C), south (S) and west (W) part of the domain. The horizontal lines indicate the 95% bootstrapped confidence interval for each vertical layer.





**Figure 5:** Seasonal average RMSE of ozone mixing ratio (ppbv) for each station and model, calculated for two height ranges: LT (lower troposphere=0-2km) and UT (upper troposphere=2-8.5km).

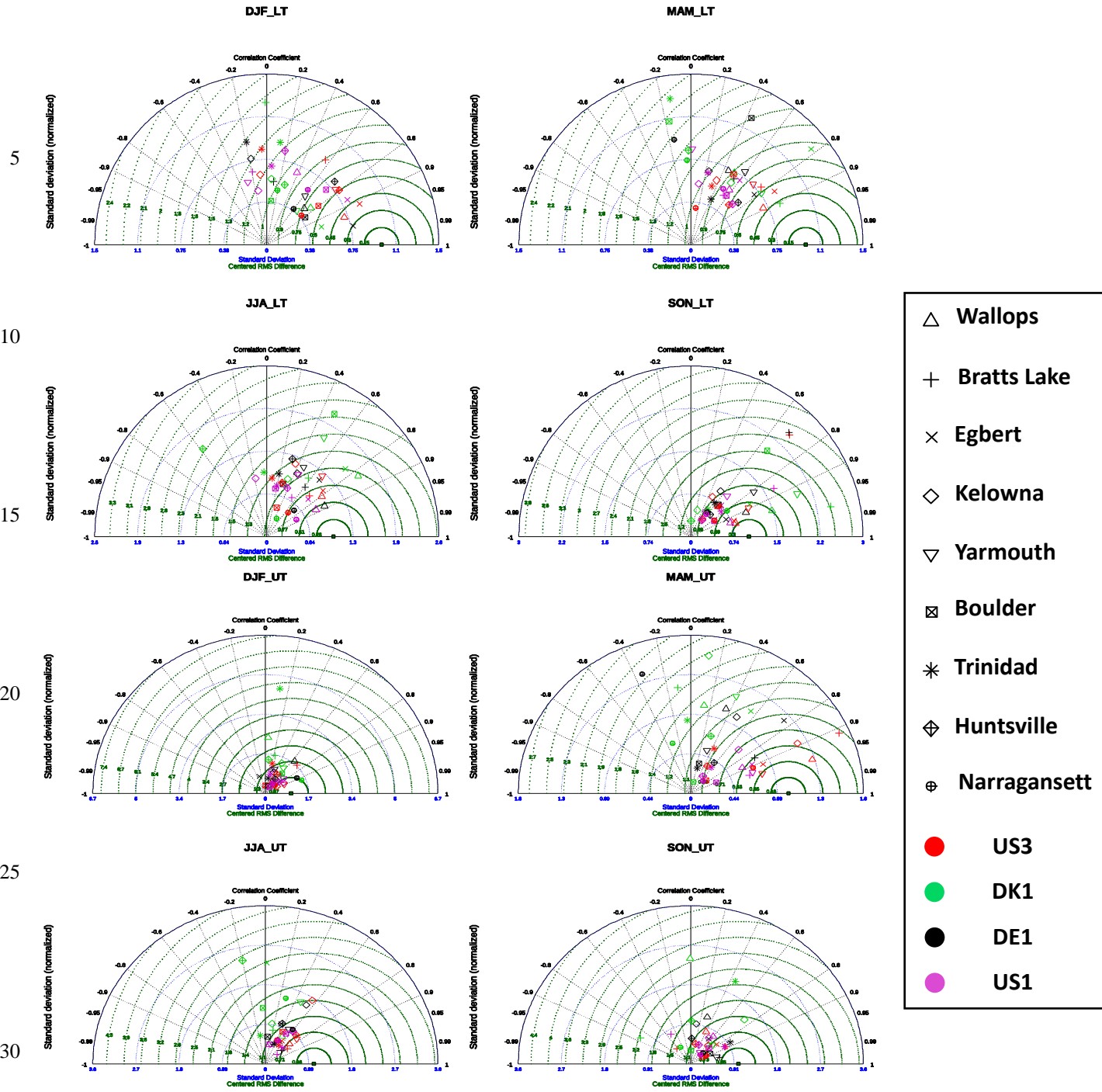

**Figure 6:** Seasonal Taylor diagrams using normalized standard deviations for two height ranges: LT (lower troposphere=0-2km) and UT (upper troposphere=2-8.5km). Stony Plain (STN021) is excluded because DE1's domain does not incorporate the site's location.

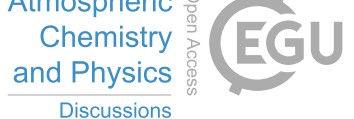





5   **Figure 7:** Average ozone profiles for winter (DJF): a) all stations, b) Northern sites, c) Western sites, d) Eastern sites.



**Figure 8:** Percentages of lateral boundary contributions (BC1, BC2 and BC3) to the total (BCtot) at each specific height
30 range, ozonesonde site, model and season. LT represents the lower troposphere (0-2km), MT the middle troposphere (2-8.5 km) and UTLS the upper troposphere to lower stratosphere (8.5-18 km). BC1=lateral boundary conditions non-zero only at the 0-750 mb level; BC2= lateral boundary conditions non-zero only at the 750-250 mb level; BC3= lateral boundary conditions are non-zero only at the levels above 250 mb.





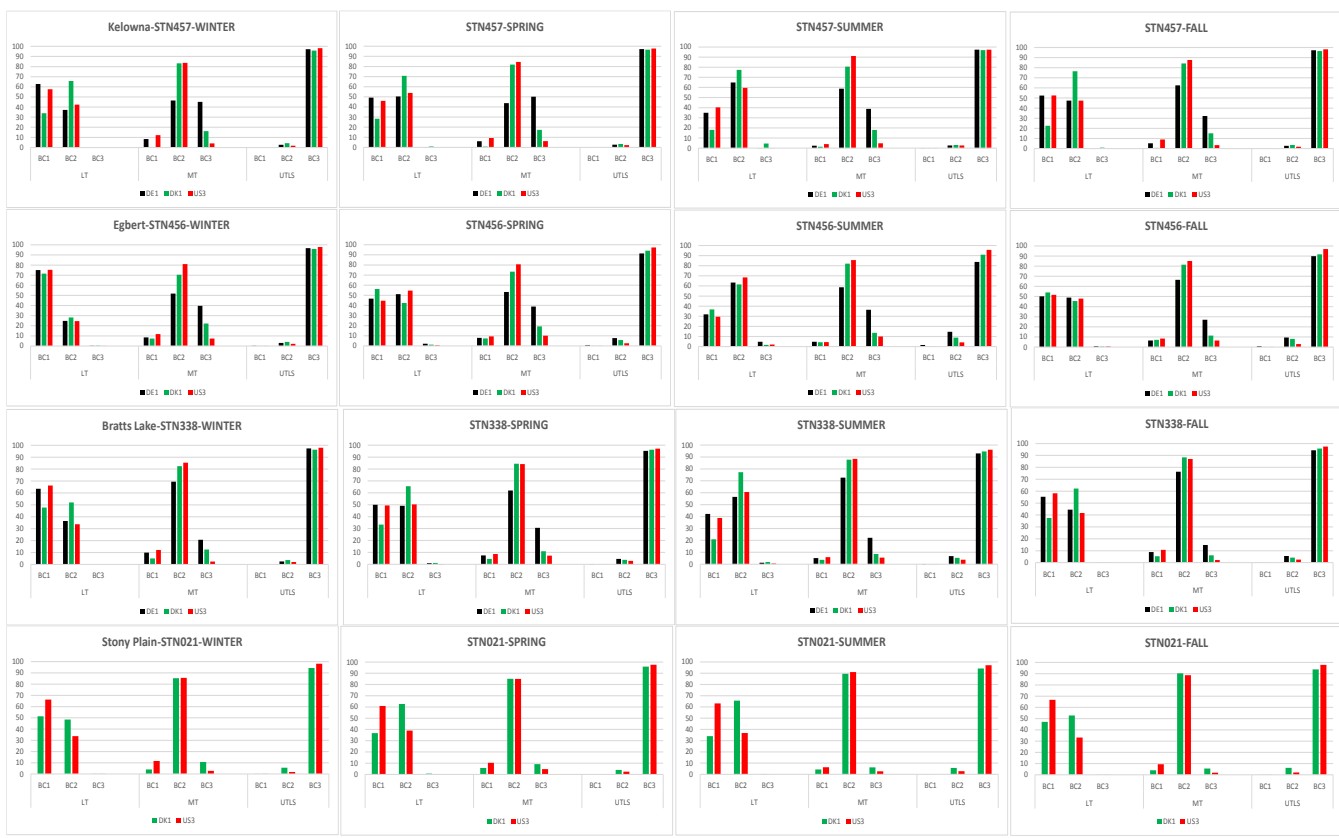

**Figure 8 (cont'd):** Percentages of lateral boundary contributions (BC1, BC2 and BC3) to the total (BCtot) at each specific height range, ozonesonde site, model and season. LT represents the lower troposphere (0-2km), MT the middle troposphere (2-8.5 km) and UTLS the upper troposphere to lower stratosphere (8.5-18 km). BC1=lateral boundary conditions non-zero only at the 0-750 mb level; BC2=lateral boundary conditions non-zero only at the 750-250 mb level; BC3=lateral boundary conditions are non-zero only at the levels above 250 mb.



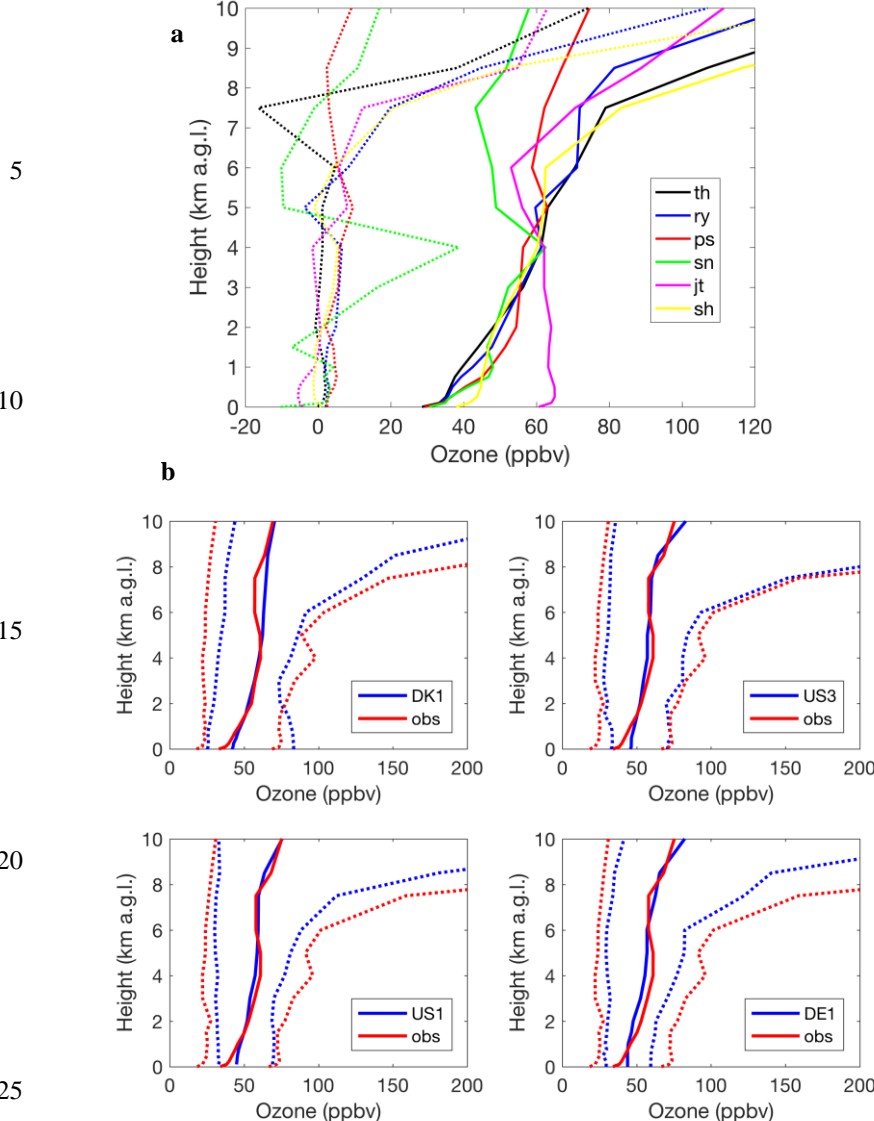

**Figure 9. a)** Mean ozone profiles using all available IONS ozonesondes at each site (10 May – 20 June 2010) interpolated at specific vertical levels. The dotted lines show the mean difference between the profiles during average and episodic conditions (episodic – average). The episodic periods taken are 22-29 May and 7-14 June. During intrusions, the average $O_3$ enhancement is up to 40 ppb in the first 8km from the surface (San Nicolas-SN; green dotted line) and reaches 105 ppb at 10km altitude (Point Reyes-RY; blue dotted line). Note that JT and SH are inland sites; all other sites are coastal. **b)** Observed (red) and modeled (blue) ozone percentiles (5th, 50th, 95th) during the May-June IONS campaign (131 profiles at 6 sites). Each panel corresponds to a different modeling system.





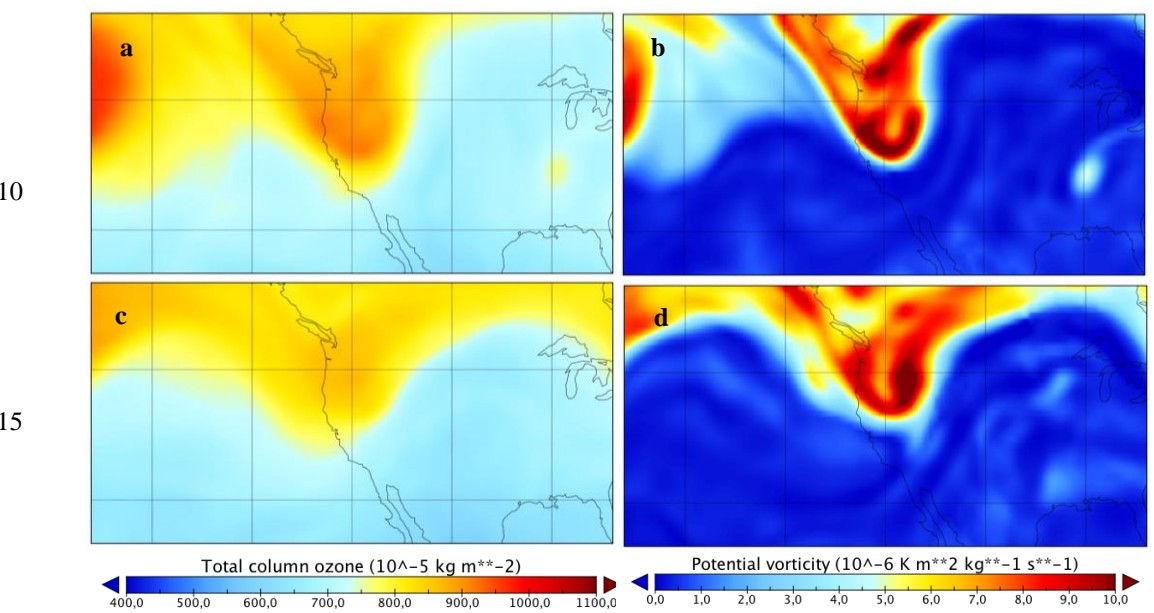

**Figure 10.** Indicative fields of total ozone column (TOC, left) and potential vorticity (IPV, right) at the 330K isentropic surface during May 28, 2010 (a and b) and June 10, 2010 (c and d). Source: Era-interim.





**Figure 11.** Ozone profiles (observed: circle; modeled: colored lines) and relative humidity (dashed line in %; shares the same scale with ozone in the x-axis) at each IONS site during the May 28 and June 8-9 intrusion. The stratospheric intrusion is denoted by the sudden drop in relative humidity that is accompanied by increase in ozone mixing ratios from the ozonesondes.



**Figure 12**. Fractional difference (%) between observed and simulated ozone profiles. Results are presented aggregated from all soundings (a, c) and at each site separately (b, d). Plots a and b use all profiles (10 May – 20 June 2010). Plots c and d present results during episodic conditions (22-29 May, 7-14 June).