# Peer review of "Seasonal ozone vertical profiles over North America using the AQMEII3 group of air quality models: model inter-comparison and stratospheric intrusions"

_Atmospheric Chemistry and Physics, 2018_

## Referee Comment (RC1) · Anonymous Referee #1 · 28 Mar 2018

**General Comments**

The AQMEII project has resulted in an unprecedented intercomparison and examination of European and North American regional air quality models. The sensitivity of regional AQ model predictions to chemical lateral boundary conditions (CLBCs) was examined in the first two phases of AQMEII. This paper extends that previous work by analyzing one-year-long North American simulations performed in AQMEII Phase 3 by four AQ modeling systems that used the same anthropogenic and wildfire emissions

and the same CLBCs. Model predictions of vertical ozone distributions across North America were compared with weekly ozonesonde data from a subset of the North American ozonesonde "network" and with a special daily ozonesonde data set from the 2010 CalNex field study in California. The comparison against measurements was also supplemented by a model-based analysis of chemically inert tracers specified at the model lateral boundaries.

As has usually been the case in AQMEII, no single model was the obvious best performer. The inert tracer analysis showed the strong dependence of interior ozone concentrations on specified CLBCs. It also appeared that model skill in predicting upper-tropospheric ozone depended more on the meteorological model used than on the AQ model used.

This is a well-written paper and I recommend acceptance of this manuscript with minor revisions. I have made a number of specific comments and suggestions below along with a number of minor editorial suggestions and corrections that I hope will improve the final version. I have also suggested revisions to a number of the figures that I believe will help readers.

**Specific Comments**

1. Since this paper is a contribution to Phase 3 of AQMEII, it would help a potential reader if the title of the paper referred to the "AQMEII-**3** group of air quality models". This would also be consistent with the use of "AQMEII-3" in the Abstract and body of the paper.

2. In the second paragraph of the Introduction the authors note the scarcity of upper-air measurements that can be used to evaluate regional AQ models. This paper makes use of ozonesonde measurements to evaluate upper-air performance, but measurements from instrumented commercial airliners from the MOZAIC program

have also been used to evaluate ozone (and NOx and CO) predictions made by regional AQ models. This second data source could be mentioned explicitly in the Introduction as part of the overview, and there are two relevant AQMEII papers that used MOZAIC data: the AQMEII-1 paper by Solazzo et al. (2013), which is referenced here, and the AQMEII-2 paper by Giordano et al. (2015, *Atmos. Environ.*, vol. 115, pp. 371-388, doi:10.1016/j.atmosenv.2015.02.034), which is not referenced but which also seems relevant.

3. I had several questions about the summary of atmospheric modeling systems provided in Section 2.1:

\* Nothing is said about the different horizontal domains considered by the four modeling systems even though the choice of domain size and location resulted in at least one model (DE1) missing one of the ozonesonde stations considered (Stony Plain). Domain extent is also not mentioned in the list of differences between modeling systems (lines 29-30).

\* Table 1 lists the horizontal grid spacing for model DK1 as 50 km but Table 1 of Solazzo et al. (2017) has a value of 16.7 km for the same model. This apparent inconsistency is more than a detail since the coarse grid spacing of DK1 is used as an explanation of model behavior (e.g., p. 6).

\* This section notes that some harmonization was achieved between modeling systems in AQMEII-3, including common simulation period, common anthropogenic and wildfire emissions, and common lateral boundary conditions. However, although the modeling domain considered includes much of Canada and part of Mexico (e.g., Figure 1), nothing is said about the Canadian and Mexican emissions that were (presumably) used.

\* On page 4, line 3 it is stated that each modeling system included 3 non-reactive tracers, but Table 1 indicates that one model (US1) did not include non-reactive tracers.
Interactive comment

4. In Section 2.3, rather than stating that the seasonality of the ozone profiles is "under-represented", is the point being made that the samples are small and hence the sampling variations may be large? And can a reference be given for the calculation of the 95% bootstrapping confidence limits?

5. I found the description of the use of the Fractional Difference Indicator in Figure 12 given at the end of Section 5 and in the Figure 12 caption to be somewhat unclear. Is FD calculated for each individual profile and then averaged or are the individual profiles averaged and then FD calculated?

6. Here are some possible changes to the figures that could make them easier to interpret and more impactful:

* Figures 2-4: Only 3 seasons are shown – fall plots are missing even though the fall period is referred to in the Fig. 2 caption. The seasonal box plots for the fall in Figure S1 are a useful supplement but not a substitute (cf. p. 6, l. 20-21).

* Figures 2-4: It would help the reader if a thicker line were used for the observed profiles. The legend and the season labels could also be made larger.

* Figure 5: The season labels are "floating" at the top of each bar chart and might appear to be associated with the bar chart above them; this confusion could be avoided if they were moved lower. Also, "Hunstv." is an odd abbreviation for Huntsville.

* Figure 6: It would help the reader if all of the labels, both on and above each Taylor diagram, could be made larger.

* Figure 7: As in Figures 2-4, use a thicker line for the observed profile. The number of sites considered in each panel could also be added in parentheses to the top label for each panel (e.g., "Winter - Western sites (2)").

* Figure 8: Why are DK1 results missing for Wallops when they are available in Figure 5? The DE1 results missing for Stony Plain, on the other hand, are expected.

* Figure 9b: Why is the observed 95th-percentile profile slightly different in the DK1 panel at about 5 km a.s.l. compared to the other three panels?

* Figure 10: The usual units for total ozone column are Dobson units. A sentence could be added to the caption to note what the TOC range shown (400 to 1100 x $10^{-5}$ kg m$^{-2}$) is in DU.

* Figure 11: If the panels are ordered from left to right by decreasing latitude, why is the Shasta panel where it is? (same question for Figure S5)

* Figures S2 and S3: Season labels could be made larger if they were moved to the right-hand side and the station names moved lower in the winter panels. The legends could also be made larger if they were moved higher up.

* Figure S3: It is stated on page 6 (line 32) that wind speed profiles are not available for Huntsville. What about Narrangansett, Boulder, and Trinidad Head, which are also not shown in this figure?

**Technical Corrections/Suggestions**

p. 1, l. 25   "... have provided *modeled* ozone vertical profiles ..." (i.e., vs. measured profiles)

p. 3, l. 18   "are used ... are performed ... are prepared"?

p. 4, l. 5-6   Lack of chemistry is mentioned several times – too repetitive?.

p. 4, l. 23   Perhaps "Ozonesonde *data* are obtained from various networks ..."

p. 4, l. 29   Perhaps "The ten sites depicted in Figure 1a had *weekly* launches throughout the entire year ..."

p. 5, l. 10   Perhaps "... at seven sites, one in southern British Columbia *(Kelowna)* and six in California"

p. 6, l. 16   "... the mean mixing ratios Narrangansett"?

p. 7, l. 12   For clarity, perhaps "For the UT, the highest errors *in ozone mixing ratio* occur during ..."

p. 7, l. 14   "Trinidad Head" rather than "Trinidad" (twice)

p. 8, l. 18   RMSD or RMSE?

p. 8, l. 30   "... but no*t* chemistry"?

p. 9, l. 5-11   The use of "mb" here is inconsistent with the use of "hPa" in the rest of the paper.

p. 9, l. 21   Liu et al. (2018)?

p. 10, l. 28   Can a reference be provided for ERA-interim?

p. 13   There are multiple reference formats being used.

p. 18   Could include characterization of the site location relative to the model domain in this table for the IONS stations (i.e., "West").

p. 30, l. 3   For the observed ozone profiles, the symbols look more like diamonds than circles.

p. 31, l. 26   Should it be "Plots a and c" and "Plots b and d"?

---

## Referee Comment (RC2) · Anonymous Referee #2 · 28 May 2018

I have read the manuscript and i would like to make the following comments:

Since the modeling systems have different horizontal grid spacing, vertical layers and meteorological drivers, can the authors connect the differences in model performance to these variations in model configuration?

Lower correlations and high UT errors are shown for spring besides winter (Figures 5 and 6). It would be helpful if the authors provide similar plots with Fig. 7 with average ozone vertical profiles from all stations for spring. This would indicate any consistent

model behavior in the vertical as with the winter case.

How are the models performing in the meteorological fields for the stratospheric intrusion cases? Can that possibly explain the underestimation of the high ozone values in the upper layers of the atmosphere?

---

## Referee Comment (RC3) · Anonymous Referee #3 · 8 Jun 2018

Review of Manuscript acp-2018-98

**Seasonal ozone vertical profiles over North America using the AQMEII group of air quality models: model inter-comparison and stratospheric intrusions**

by Marina Astitha et al.

June 6, 2018

The manuscript provides multi-model simulations of ozone profiles for a number of observational sites in the United States and Europe and validates the model results. The models seem to underestimate ozone up to 6 km. For stratospheric intrusions, the ozone maxima are also underestimated between 2 and 6 km.

It is difficult to judge where the advances of this study are. There have been numerous modelling efforts for evaluating the ozone budget in the more recent past such as (Stevenson et al., 2006), (Wild, 2007), (Young et al., 2013), or (Knowland et al., 2017). None of these papers are cited or included in the discussion. Spatial resolution is an important issue (e.g., Roelofs et al., 2003; Eastman and Jacob, 2016), and at least a good horizontal resolution of 0.25º × 0.25º is reported. However, no information on the vertical resolution is given in Sec. 2.1. In Sec. 2.2 an interpolation to 18 "standard vertical heights" up to 18000 m is mentioned. This kind of grid does not allow one to resolve narrow atmospheric layers. For this reason, also the value of the figures shown is limited. There is a host of literature on stratosphere-to-troposphere transport after the 2003 review by Stohl et al., in particular from North America, Europe and East Asia (have a look at papers citing the review paper!), also discussing the role of mixing (Trickl et al., 2014; 2016).

In summary, I cannot recommend publishing this manuscript in the current version.

References:

Eastman, S. D., and Jacob, D. J.: Limits on the ability of global Eulerian models to resolve intercontinental transport of chemical plumes, Atmos. Chem. Phys., 17, 2543-2553, 2017.

Knowland, K., E., Doherty, R. M., Hodges, K. I., and Ott, L. E.: The influence of mid-latitude cyclones on European background surface ozone, Atmos. Chem. Phys., 17, 12421-12447, 2017.

Roelofs, G. J., Kentarchos, A. S., Trickl, T., Stohl, A., Collins, W. J., Crowther, R. A., Hauglustaine, D., Klonecki, A., Law, K. S., Lawrence, M. G., von Kuhlmann, R., and van Weele, M.: Intercomparison of tropospheric ozone models: Ozone transport in a complex tropopause folding event, J. Geophys. Res. 108, 8529, doi:10.1029/2003JD003462, STA 14, 13 pp., 2003.

Stevenson, D. S., Dentener, F. J., Schultz, M. G., Ellingsen, K., van Noije, T. P. C., Wild, O., Zeng, G., Amann, M., Atherton, C. S., Bell, N., Bergmann, D. J., Bey, I., Butler, T., Cofala, J., Collins, W. J., Derwent, R. G., Doherty, R,. M., Drevet, J., Eskes, H. J., Fiore, A. M., Gauss, M,., Hauglustaine, D. A., Horowitz, L. W., Isaksen, I. S. A., Krol, M. C., Lamarque, J.-F., Lawrence, M. G., Montanaro, V., Müller, J.-F., Pitari, G., Prather, M. J., Pyle, J. A., Rast, S., Rodriguez, J. M., Sanderson, M. G., Savage, N. H., Shindell, D. T., Strahan, S. E., Sudo, K., and Szopa, S.: Multimodel ensemble simulations of present-day and near-future tropospheric ozone, J. Geophys. Res., 111, D08301, doi: 10.1029/2005JD006338, 23 pp., 2006.

Trickl, T., Vogelmann, H., Giehl, H., Scheel, H. E., Sprenger, M., and Stohl, A.: How stratospheric are deep stratospheric intrusions? Atmos. Chem. Phys., 14, 9941-9961, 2014.

Trickl, T., Vogelmann, H., Fix, A., Schäfler, A., Wirth, M., Calpini, B., Levrat, G., Romanens, G., Apituley, A., Wilson, K. M., Begbie, R., Reichardt, J., Vömel, H. and Sprenger, M.: How stratospheric are deep stratospheric intrusions into the troposphere? LUAMI 2008, Atmos. Chem. Phys, 16, 8791-8815, 2016.

Wild, O.: Modelling the global tropospheric ozone budget: exploring the variability in current models, Atmos. Chem. Phys., 7, 2643–2660, 2007.

Young, P. J., Archibald A. T., Bowman, K. W., Lamarque, J.-F., Naik, V., Stevenson, D. S., Tilmes, S., Voulgarakis, A., Wild, O., Bergmann, D., Cameron-Smith, P., Cionni, I., Collins, W. J., Dalsøren, S. B., Doherty, R. M., Eyring, V., Faluvegi, G., Horowitz, L.W., Josse, B., Leen, Y. H., MacKenzie, I. A., Nagashima, T., Plummer, D. A., Righi1, M., Rumbold, S. T., Skeie, R. B., Shindell, D. T., Strode, S. A., Sudo, K., Szopa, S., and Zeng, G..: Pre-industrial to end 21st century projections of tropospheric ozone from the Atmospheric Chemistry and Climate Model Intercomparison Project (ACCMIP), Atmos. Chem. Phys., 13, 2063–2090, 2013.

---

## Author Comment (AC1) · 4 Aug 2018

**REVIEWER#1**

**General Comments**

The AQMEII project has resulted in an unprecedented intercomparison and examination of European and North American regional air quality models. The sensitivity of regional AQ model predictions to chemical lateral boundary conditions (CLBCs) was examined in the first two phases of AQMEII. This paper extends that previous work by analyzing one-year-long North American simulations performed in AQMEII Phase 3 by four AQ modeling systems that used the same anthropogenic and wildfire emissions and the same CLBCs. Model predictions of vertical ozone distributions across North America were compared with weekly ozonesonde data from a subset of the North American ozonesonde "network" and with a special daily ozonesonde data set from the 2010 CalNex field study in California. The comparison against measurements was also supplemented by a model-based analysis of chemically inert tracers specified at the model lateral boundaries.

As has usually been the case in AQMEII, no single model was the obvious best performer. The inert tracer analysis showed the strong dependence of interior ozone concentrations on specified CLBCs. It also appeared that model skill in predicting upper-tropospheric ozone depended more on the meteorological model used than on the AQ model used.

This is a well-written paper and I recommend acceptance of this manuscript with minor revisions. I have made a number of specific comments and suggestions below along with a number of minor editorial suggestions and corrections that I hope will improve the final version. I have also suggested revisions to a number of the figures that I believe will help readers.

AUTHORS' RESPONSE: We would like to thank the reviewer for all comments and suggestions. We carefully addressed the comments and clarified obscure parts that we believe have improved the coherence of the manuscript.

Please find detailed responses to each comment below. Reviewer' comments are in black, authors' replies in blue and revised sentences in italics. We also provide a version of the manuscript with "Track Changes" where all changes and additions are easily discernible.

**Specific Comments**

1. Since this paper is a contribution to Phase 3 of AQMEII, it would help a potential reader if the title of the paper referred to the "AQMEII-3 group of air quality models". This would also be consistent with the use of "AQMEII-3" in the Abstract and body of the paper.

Authors' Response: Amended.

2. In the second paragraph of the Introduction the authors note the scarcity of upper-air measurements that can be used to evaluate regional AQ models. This paper makes use of ozonesonde measurements to evaluate upper-air performance, but measurements from instrumented commercial airliners from the MOZAIC program have also been used to evaluate ozone (and NOx and CO) predictions made by regional AQ models. This second data source could be mentioned explicitly in the Introduction as part of the overview, and there are two relevant AQMEII papers that used MOZAIC data: the AQMEII-1 paper by Solazzo et al. (2013), which is referenced here, and the AQMEII-2 paper by Giordano et al. (2015, Atmos. Environ., vol. 115, pp. 371-388, doi:10.1016/j.atmosenv.2015.02.034), which is not referenced but which also seems relevant.

Authors' Response: We added a reference to the usage of MOZAIC datasets as well as the Giordano et al. (2015) paper in the 2nd paragraph of the Introduction.

3. I had several questions about the summary of atmospheric modeling systems provided in Section 2.1:

* Nothing is said about the different horizontal domains considered by the four modeling systems even though the choice of domain size and location resulted in at least one model (DE1) missing one of the ozonesonde stations considered (Stony Plain). Domain extent is also not mentioned in the list of differences between modeling systems (lines 29-30).

Authors' Response: All modeling groups were required to configure their models to a North American common analysis domain with the following extent: 130°W to 59.5°W, 23.5°N to 58.5°N. Differences in the geographic projection resulted in some discrepancies on the sites included in the domain. We added that information in Table 1.

* Table 1 lists the horizontal grid spacing for model DK1 as 50 km but Table 1 of Solazzo et al. (2017) has a value of 16.7 km for the same model. This apparent inconsistency is more than a detail since the coarse grid spacing of DK1 is used as an explanation of model behavior (e.g., p. 6).

Authors' Response: We thank the reviewer for noticing this mistake. The table and model performance evaluation are corrected in the revised version of the manuscript (please see the revised version with all corrections visible through the "Track changes" option).

* This section notes that some harmonization was achieved between modeling systems in AQMEII-3, including common simulation period, common anthropogenic and wildfire emissions, and common lateral boundary conditions. However, although the modeling domain considered includes much of Canada and part of Mexico (e.g., Figure 1), nothing is said about the Canadian and Mexican emissions that were (presumably) used.

Authors' Response: Canadian emissions were provided by Environment Canada. Emissions for Mexico are based on a 1999 inventory, using population as a surrogate for spatial allocation. The emission inventory for North America used in the AQMEII3 simulations is described in detail in Pouliot et al. 2015. We added this information in the manuscript (section 2.1).

* On page 4, line 3 it is stated that each modeling system included 3 non-reactive tracers, but Table 1 indicates that one model (US1) did not include non-reactive tracers.

Authors' Response: All groups used tracers but three of the four groups provided 3D outputs of tracer concentrations that are used in the analysis. The wording in this sentence is corrected to be in accordance with Table 1.

4. In Section 2.3, rather than stating that the seasonality of the ozone profiles is "underrepresented", is the point being made that the samples are small and hence the sampling variations may be large? And can a reference be given for the calculation of the 95% bootstrapping confidence limits?

Authors' Response: We agree with the reviewer's comment that the sample size can influence the variability rather than the seasonality of ozone profiles. We revised the wording in section 2.3 to "*The modeled ozone mixing ratios are sampled in accordance to available ozonosondes, thus the variability of the vertical ozone profiles might be under-represented since the ozonesondes are not continuous*

*throughout each month (Lin et al. 2015)".* A reference for the bootstrapping approach is given in the text (Efron, 1987).

5. I found the description of the use of the Fractional Difference Indicator in Figure 12 given at the end of Section 5 and in the Figure 12 caption to be somewhat unclear. Is FD calculated for each individual profile and then averaged or are the individual profiles averaged and then FD calculated?

Authors' Response: The first option is true. An explanation is added to clarify the case.

6. Here are some possible changes to the figures that could make them easier to interpret and more impactful:

* Figures 2-4: Only 3 seasons are shown – fall plots are missing even though the fall period is referred to in the Fig. 2 caption. The seasonal box plots for the fall in Figure S1 are a useful supplement but not a substitute (cf. p. 6, l. 20-21).

Authors' Response: We excluded the vertical profiles for the fall season because there is less variation between model outputs and observations. That helped to make the plots easier to read. Nevertheless, we included the fall vertical profiles in the supplement as suggested by the reviewer (new Fig. S1).

* Figures 2-4: It would help the reader if a thicker line were used for the observed profiles. The legend and the season labels could also be made larger.

Authors' Response: We tried to make the observed lines thicker but it did not improve the plots, mainly because the lines are very close together. A larger legend is included in the 3rd column of figures 2-4. The seasons are now mentioned in the caption. We produced the plots with high resolution that does not alter the quality of the figure when zooming in to see the details. We believe this is also helpful for the readers.

* Figure 5: The season labels are "floating" at the top of each bar chart and might appear to be associated with the bar chart above them; this confusion could be avoided if they were moved lower. Also, "Hunstv." is an odd abbreviation for Huntsville.

Authors' Response: The figure is now revised according to the reviewer's suggestions to make the season labels clear. The abbreviation for Huntsville is corrected to "Hunts".

* Figure 6: It would help the reader if all of the labels, both on and above each Taylor diagram, could be made larger.

Authors' Response: We produced all plots in vector format to achieve high resolution quality when zoomed in and increased the font size as suggested. We will work with the editorial requirements for additional improvement of the plots before the manuscript is published in its final form.

* Figure 7: As in Figures 2-4, use a thicker line for the observed profile. The number of sites considered in each panel could also be added in parentheses to the top label for each panel (e.g., "Winter - Western sites (2)").

Authors' Response: Amended.

* Figure 8: Why are DK1 results missing for Wallops when they are available in Figure 5? The DE1 results missing for Stony Plain, on the other hand, are expected.

Authors' Response: Figure 8 shows results for the tracer concentrations (BC1, 2 and 3) which are different from the RMSE results for ozone mixing ratios in Fig.5. The data was directly uploaded to the JRC Ensembles system by each modeling group which we used to conduct the analysis in this study.

* Figure 9b: Why is the observed 95th-percentile profile slightly different in the DK1 panel at about 5 km a.s.l. compared to the other three panels?

Authors' Response: Figure 9b is revised.

* Figure 10: The usual units for total ozone column are Dobson units. A sentence could be added to the caption to note what the TOC range shown (400 to 1100 x 10−5 kg m−2) is in DU.

Authors' Response: The caption of Figure 10 is revised to:

*Figure 10. Indicative fields of total ozone column (TOC, left) and potential vorticity (IPV, right) at the 330K isentropic surface during May 28, 2010 (a and b) and June 10, 2010 (c and d). The colorbar ranges correspond to Dobson units (DU) and Potential Vorticity units (PVU) respectively. Source: Era-interim.*

* Figure 11: If the panels are ordered from left to right by decreasing latitude, why is the Shasta panel where it is? (same question for Figure S5)

Authors' Response: Figures 11 and S6 (previously S5) are revised to reflect the correct location of the sites.

* Figures S2 and S3: Season labels could be made larger if they were moved to the right-hand side and the station names moved lower in the winter panels. The legends could also be made larger if they were moved higher up.

Authors' Response: To improve the readability of the figures (now S3 and S4) we increased the size of the plots, added information on the panels in the caption and increased the size of the legends.

* Figure S3: It is stated on page 6 (line 32) that wind speed profiles are not available for Huntsville. What about Narrangansett, Boulder, and Trinidad Head, which are also not shown in this figure?

Authors' Response: We mention the lack of wind speed observations for Huntsville in that section to explain that we could not further examine the large RMSE of DK1 seen for Huntsville. We revised the sentence to avoid any confusion. We also added information of wind and temperature data availability in section 2.2.

**Technical Corrections/Suggestions**

p. 1, l. 25 "... have provided modeled ozone vertical profiles ..." (i.e., vs. measured profiles)

Authors' Response: The word "modeled" is added in the sentence accordingly.

p. 3, l. 18 "are used ... are performed ... are prepared"?

Authors' Response: the sentence is revised.

p. 4, l. 5-6 Lack of chemistry is mentioned several times – too repetitive?.

Authors' Response: The tracer experiments (where the lack of chemistry is important) are described for the first time in page 4. We removed the repetition in section 4 (discussion of the tracer results).

p. 4, l. 23 Perhaps "Ozonesonde data are obtained from various networks ..."

Authors' Response: Amended.

p. 4, l. 29 Perhaps "The ten sites depicted in Figure 1a had weekly launches throughout the entire year ..."

Authors' Response: The launches are not weekly throughout the year for all stations.

p. 5, l. 10 Perhaps "... at seven sites, one in southern British Columbia (Kelowna) and six in California"

Authors' Response: Amended.

p. 6, l. 16 "... the mean mixing ratios Narrangansett"?

Authors' Response: Corrected.

p. 7, l. 12 For clarity, perhaps "For the UT, the highest errors in ozone mixing ratio occur during ..."

Authors' Response: Amended.

p. 7, l. 14 "Trinidad Head" rather than "Trinidad" (twice)

Authors' Response: Amended

p. 8, l. 18 RMSD or RMSE?

Authors' Response: Corrected (it is RMSE).

p. 8, l. 30 "... but not chemistry"?

Authors' Response: the sentence is removed to avoid repetition.

p. 9, l. 5-11 The use of "mb" here is inconsistent with the use of "hPa" in the rest of the paper.

Authors' Response: Corrected.

p. 9, l. 21 Liu et al. (2018)?

Authors' Response: Corrected.

p. 10, l. 28 Can a reference be provided for ERA-interim?

Authors' Response: Amended.

p. 13 There are multiple reference formats being used.

Authors' Response: Amended.

p. 18 Could include characterization of the site location relative to the model domain in this table for the IONS stations (i.e., "West").

Authors' Response: Amended.

p. 30, l. 3 For the observed ozone profiles, the symbols look more like diamonds than circles.

Authors' Response: Amended.

p. 31, l. 26 Should it be "Plots a and c" and "Plots b and d"?

Authors' Response: The description of figure caption 12 is correct. Plots a and b are constructed using all available profiles, while c and d depict the episodic conditions only.

---

## Author Comment (AC2) · 4 Aug 2018

**REVIEWER#2**

I have read the manuscript and i would like to make the following comments:

AUTHORS' RESPONSE: We would like to thank the reviewer for all comments and suggestions. We carefully addressed the comments and clarified obscure parts that we believe have improved the coherence of the manuscript.

Please find detailed responses to each comment below. Reviewer' comments are in black, authors' replies in blue and revised sentences in italics. We also provide a version of the manuscript with "Track Changes" where all changes and additions are easily discernible.

Since the modeling systems have different horizontal grid spacing, vertical layers and meteorological drivers, can the authors connect the differences in model performance to these variations in model configuration?
Authors' Response: This is a very interesting question. Through this investigation, no specific model was the "winner" in terms of performance. There was no apparent benefit that depended on grid spacing. The only thing that came up more frequently was that the meteorological driver plays more important role than the air quality model, without having indications of a "best" model among the participants. We expanded on this topic in the conclusions of the revised manuscript.

Lower correlations and high UT errors are shown for spring besides winter (Figures 5 and 6). It would be helpful if the authors provide similar plots with Fig. 7 with average ozone vertical profiles from all stations for spring. This would indicate any consistent model behavior in the vertical as with the winter case.
Authors' Response: Two new panels are added in Figure 7 with average ozone profiles from all stations for spring and summer. Discussion about the added profiles is included in section 3 of the revised manuscript.

How are the models performing in the meteorological fields for the stratospheric intrusion cases? Can that possibly explain the underestimation of the high ozone values in the upper layers of the atmosphere?
Authors' Response: Surface temperature, wind speed and wind direction and vertical profiles of temperature and wind speed for some stations in North America have been evaluated in the work of Solazzo et al. 2017. According to Solazzo et al. (2017), model performance in meteorological fields showed that temperature and wind speed were biased low and high. However, the differences in location of the sites used in this study compared to the ones in Solazzo et al. (2017) do not allow for a generalization of the conclusions. Our comparison of wind speed and temperature (Figures S3 and S4) are not conclusive on the type of influence exerted to the ozone vertical profiles from meteorological fields due to the inherent nonlinearity among physical and chemical processes. We added a new figure in the supplement that shows vertical profiles of ozone and potential temperature for Point Reyes for each modeling system (Fig. S7). A discussion on this issue is added in section 5.

---

## Author Comment (AC3) · 4 Aug 2018

Please see responses to all comments in the Supplement

Please also note the supplement to this comment:
https://www.atmos-chem-phys-discuss.net/acp-2018-98/acp-2018-98-AC3-supplement.pdf
* * *
[Figure]

2018.

---

## Author Comment (AC4) · 4 Aug 2018

**REVIEWER#3**

The manuscript provides multi-model simulations of ozone profiles for a number of observational sites in the United States and Europe and validates the model results. The models seem to underestimate ozone up to 6 km. For stratospheric intrusions, the ozone maxima are also underestimated between 2 and 6 km.

It is difficult to judge where the advances of this study are. There have been numerous modelling efforts for evaluating the ozone budget in the more recent past such as (Stevenson et al., 2006), (Wild, 2007), (Young et al., 2013), or (Knowland et al., 2017). None of these papers are cited or included in the discussion. Spatial resolution is an important issue (e.g., Roelofs et al., 2003; Eastman and Jacob, 2016), and at least a good horizontal resolution of 0.25º × 0.25º is reported. However, no information on the vertical resolution is given in Sec. 2.1. In Sec. 2.2 an interpolation to 18 "standard vertical heights" up to 18000 m is mentioned. This kind of grid does not allow one to resolve narrow atmospheric layers. For this reason, also the value of the figures shown is limited. There is a host of literature on stratosphere-to-troposphere transport after the 2003 review by Stohl et al., in particular from North America, Europe and East Asia (have a look at papers citing the review paper!), also discussing the role of mixing (Trickl et al., 2014; 2016).

In summary, I cannot recommend publishing this manuscript in the current version.

Authors' Response: We would like to thank the reviewer for providing these references. However, we need to note that this study did not include any global atmospheric chemistry models neither a European domain. All of the mentioned publications discuss global modeling of ozone (tropospheric and stratospheric), primarily for Europe. Our study focuses on vertical ozone distribution and stratospheric intrusions over North America, with four research groups from the US and Europe providing year-long regional scale simulations (Fig. 1a). The horizontal grid spacing of all modeling systems ranges from 12 to 24 km (Table 1) and the vertical grid spacing varies depending on the model (information on the vertical resolution is included in Table 1). The 18 standard vertical layers mentioned in the manuscript are used to compare model outputs with ozonesonde data as these layers align better with the ozonesonde launches. Model data was interpolated from each native model grid to those 18 layers.

The vertical resolution of each model is given in a recent publication of Liu et al. (2018) and shown below:

[Figure]

AQMEII Phase 3 (AQMEII3) is devoted to performing joint modelling experiments with HTAP2. The AQMEII modelling community (Table 5) includes almost all of the major existing modelling systems for regional-scale chemical transport simulation in research and regulatory applications on both continents. Most of the groups participating are part of modelling initiatives in the individual European member states, and some of these groups utilize models developed in North America, thus providing the opportunity of assessing the application of these models outside of their conventional modelling context.

The unique configuration of the model simulations conducted under AQMEII3 is that "*common anthropogenic emission inventories and lateral chemical boundary conditions were implemented by all modeling groups, which helps us further investigate model-to-model variability and performance evaluation for the vertical distribution of ozone mixing ratios.*" Our contribution to the scientific knowledge of modeled ozone vertical profiles in the regional scale is new information about how different meteorological drivers, air quality models, grid resolution, and lateral boundary conditions influence the seasonal depiction of ozone vertical profiles. This information can help model developers improve model performance by looking at specific processes and configurations.

References:

Eastman, S. D., and Jacob, D. J.: Limits on the ability of global Eulerian models to resolve intercontinental transport of chemical plumes, Atmos. Chem. Phys., 17, 2543-2553, 2017.

Knowland, K., E., Doherty, R. M., Hodges, K. I., and Ott, L. E.: The influence of mid-latitude cyclones on European background surface ozone, Atmos. Chem. Phys., 17, 12421-12447, 2017.

Roelofs, G. J., Kentarchos, A. S., Trickl, T., Stohl, A., Collins, W. J., Crowther, R. A., Hauglustaine, D., Klonecki, A., Law, K. S., Lawrence, M. G., von Kuhlmann, R., and van Weele, M.: Intercomparison of tropospheric ozone models: Ozone transport in a complex tropopause folding event, J. Geophys. Res. 108, 8529, doi:10.1029/2003JD003462, STA 14, 13 pp., 2003.

Stevenson, D. S., Dentener, F. J., Schultz, M. G., Ellingsen, K., van Noije, T. P. C., Wild, O., Zeng, G., Amann, M., Atherton, C. S., Bell, N., Bergmann, D. J., Bey, I., Butler, T., Cofala, J., Collins, W. J., Derwent, R. G., Doherty, R,. M., Drevet, J., Eskes, H. J., Fiore, A. M., Gauss, M,.,Hauglustaine, D. A., Horowitz, L. W., Isaksen, I. S. A., Krol, M. C., Lamarque, J.-F., Lawrence, M.G., Montanaro, V., Müller, J.-F., Pitari, G., Prather, M. J., Pyle, J. A., Rast, S., Rodriguez, J. M.,Sanderson, M. G., Savage, N. H., Shindell, D. T., Strahan, S. E., Sudo, K., and Szopa, S.: Multimodel ensemble simulations of present-day and near-future tropospheric ozone, J. Geophys. Res., 111, D08301, doi: 10.1029/2005JD006338, 23 pp., 2006.

Trickl, T., Vogelmann, H., Giehl, H., Scheel, H. E., Sprenger, M., and Stohl, A.: How stratospheric are deep stratospheric intrusions? Atmos. Chem. Phys., 14, 9941-9961, 2014.

Trickl, T., Vogelmann, H., Fix, A., Schäfler, A., Wirth, M., Calpini, B., Levrat, G., Romanens, G., Apituley, A., Wilson, K. M., Begbie, R., Reichardt, J., Vömel, H. and Sprenger, M.: How stratospheric are deep stratospheric intrusions into the troposphere? LUAMI 2008, Atmos. Chem. Phys, 16, 8791-8815, 2016.

Wild, O.: Modelling the global tropospheric ozone budget: exploring the variability in current models, Atmos. Chem. Phys., 7, 2643–2660, 2007.

Young, P. J., Archibald A. T., Bowman, K. W., Lamarque, J.-F., Naik, V., Stevenson, D. S., Tilmes, S., Voulgarakis, A., Wild, O., Bergmann, D., Cameron-Smith, P., Cionni, I., Collins, W. J., Dalsøren, S. B.,

Doherty, R. M., Eyring, V., Faluvegi, G., Horowitz, L.W., Josse, B., Leen, Y. H., MacKenzie, I. A., Nagashima, T., Plummer, D. A., Righi1, M., Rumbold, S. T., Skeie, R. B., Shindell, D. T., Strode, S. A., Sudo, K., Szopa, S., and Zeng, G..: Pre-industrial to end 21st century projections of tropospheric ozone from the Atmospheric Chemistry and Climate Model Intercomparison Project (ACCMIP), Atmos. Chem. Phys., 13, 2063–2090, 2013.